# Head-to-Head Comparison of [18F]F-choline and Imaging of Prostate-Specific Membrane Antigen, Using [18F]DCFPyL PET/CT, in Patients with Biochemical Recurrence of Prostate Cancer

Laura García-Zoghby [1], Cristina Lucas-Lucas [2], Mariano Amo-Salas [3], Ángel María Soriano-Castrejón [1] and Ana María García-Vicente [1,*]

[1] Nuclear Medicine Department, University Hospital of Toledo, 45007 Toledo, Spain; lgarciaz@sescam.jccm.es (L.G.-Z.); asoriano@sescam.org (Á.M.S.-C.)
[2] Nuclear Medicine Department, University General Hospital of Ciudad Real, 13005 Ciudad Real, Spain; clucasl@sescam.jccm.es
[3] Department of Mathematics, Castilla-La Mancha University, 13071 Ciudad Real, Spain; mariano.amo@uclm.es
* Correspondence: amgarcia@sescam.jccm.es; Tel./Fax: +34-925-269200

**Abstract:** Purpose: To analyse diagnostic and therapeutic impact of molecular imaging TNM (miTNM) stage obtained with [18F]DCFPyL versus [18F]F-choline in head-to-head comparison in biochemical recurrence (BCR) of prostate cancer (PCa). Material and methods: Patients with BCR of PCa after radical treatment with previous [18F]F-choline-PET/CT (negative or oligometastatic disease) were recruited to [18F]DCFPyL-PET/CT. Patients were classified according to: grade group, European Association of Urology classification, PSA, PSA doubling time (PSAdt) and PSA velocity (PSAvel). The overall detection rate (DR) and miTNM stage according to PROMISE criteria were assessed for both radiotracers and also correlated (Kappa). The influence of PSA and kinetics on both PET/CT (DR and miTNM) and predictive value of unfavourable kinetics on miTNM were determined. Cut-off PSA, PSAdt and PSAvel values able to predict PET/CT results were determined. Change in miTNM and treatment derived from [18F]DCFPyL information compared with [18F]F-choline were also evaluated. Results: We studied 138 patients. [18F]DCFPyL showed a higher DR than [18F]F-choline (64.5% versus 33.3%) with a fair agreement. [18F]DCFPyL and [18F]F-choline detected T in 33.3% versus 19.6%, N in 27.5% versus 13.8%, and M in 30.4% versus 8.7%. Both tracers' DR showed significant associations with PSA and PSAvel. Significant association was only found between miTNM and PSA on [18F]F-choline-PET/CT (*p* = 0.033). For [18F]F-choline and [18F]DCFPyL-PET/CT, a PSAdt cut-off of 4.09 and 5.59 months, respectively, were able to predict M stage. [18F]DCFPyL changed therapeutic management in 40/138 patients. Conclusions: [18F]DCFPyL provides a higher DR and superior miTNM staging than [18F]F-choline in restaging BCR, especially with high PSA and unfavourable PSA kinetics, showing a fair agreement to [18F]F-choline.

**Keywords:** [18F]DCFPyL; [18F]F-choline; miTNM; PSA level; PSA kinetics; therapeutic impact

## 1. Introduction

Up to a half of pT2-3 node-negative prostate cancer (PCa) patients experience biochemical recurrence (BCR) after radical prostatectomy (RP) or radiotherapy [1]. Detection of responsible lesions in the context of a BCR constitutes a major challenge for conventional imaging modalities such as computed tomography (CT) and bone scan.

PET/CT with choline-based tracers has been the traditional imaging modality of choice in restaging patients following BCR [2]. However, multiple studies have shown low sensitivity and specificity, particularly at low prostate-specific antigen (PSA) levels, which can result in delays in salvage therapies [3,4]. For several years, and due to these

limitations, the development of radionuclides that recognizes prostate-specific membrane antigen (PSMA) ligands has been proposed as an alternative, with higher sensitivity and specificity in BCR of PCa [5]. These "top diagnostic" radiotracers have increased the detection rate (DR) of oligometastatic disease that has driven recent advancements in metastasis-directed treatment strategies.

[$^{18}$F]DCFPyL[2-(3-(1-carboxy-5-[(6-[$^{18}$F]fluoro-pyridine-3-carbonyl)-amino]-pentyl)-ureido)-pentanedioic acid] is a radiofluorinated, small-molecule, high-affinity inhibitor of PSMA [6]. The current restrictions in its use in our environment explain the dual-tracer diagnostic approach in some cases of BCR, especially in those with a PSA level >2ng/mL and a previous negative or ambiguous PET/CT with choline-based tracers. Some studies have addressed utility of 68Ga-tracers PSMA-targeting radiopharmaceuticals and choline-based tracers in head-to-head comparison [7,8], although no previous reported experience exists using the newest developed [$^{18}$F]DCFPyL. On the other hand, if we only use DR to compare both radiotracers, the real diagnostic potential of PSMA-targeting radiopharmaceuticals compared with choline-based tracers may be limited. In addition, differences in therapeutic impact have been scarcely assessed [9].

The Prostate Cancer Molecular Imaging Standardized Evaluation (PROMISE) criteria summarize standards for study design and reporting of PCa molecular imaging. PROMISE criteria propose a molecular imaging TNM (miTNM) for the interpretation of PSMA-targeting radiopharmaceuticals PET/CT designed to organize findings in comprehensible categories and to promote the exchange of information among physicians and institutions [10].

The aim of our study was multiple: (i) to analyse the concordance between [$^{18}$F]DCFPyL and [$^{18}$F]F-choline, in head-to-head comparison, regarding DR and miTNM stage using PROMISE criteria, (ii) to address the predictive value of unfavourable PSA kinetics on miTNM, and (iii) to assess the therapeutic impact of [$^{18}$F]DCFPyL compared with [$^{18}$F]F-choline-PET/CT in patients with BCR of PCa.

## 2. Material and Methods

### 2.1. Patients

Patients with BCR of PCa after radical treatment (RP, radiotherapy or both) were recruited from different hospitals of our region for re-staging with [$^{18}$F]F-choline-PET/CT between August 2020 and December 2021. No patient was under androgen deprivation therapy (ADT). Patients with negative or ambiguous [$^{18}$F]F-choline-PET/CT, or with oligometastatic disease, underwent [$^{18}$F]DCFPyL-PET/CT and were included in a prospective dataset. We established as oligometastatic disease if there is a presence of $\leq$ 3 lesions affecting lymph node (pelvis and/or retroperitoneum) or bone.

[$^{18}$F]DCFPyL-PET/CT was performed within the context of compassionate use under the approval of the Spanish Agency of Medication and Health Care Products and after being approved by a multidisciplinary committee and after receiving patients' informed and signed consent. Database registry analysis of patients was approved by an Ethical Committee (internal code of 2022-53).

The inclusion criteria for the present analysis: (i) time window between both PET/CT within 2 months and (ii) minimum clinical follow-up of 6 months.

Patients were classified in groups taking into account: grade group (1 to 5) [11], European Association of Urology (EAU) classification adapted from D'Amico risk category (low/intermediate/high) [1], PSA value closest to PET/CTs (PSA $\leq$ 1 ng/mL, 1 < PSA $\leq$ 2 and PSA > 2), PSA doubling time (PSAdt)$\leq$ or >6 months and PSA velocity (PSAvel)$\geq$ or <0.2 ng/mL/month. The initial radical treatment and subsequent salvage treatment, if previous BCR, were obtained.

### 2.2. Acquisition Protocol

[$^{18}$F]F-choline and [$^{18}$F]DCFPyL PET/CTs were performed in a unique reference hospital and with the same hybrid PET/CT scanner (Discovery 5R/IQ, GE) in 3D acquisition

mode for 2 min per bed position. Low dose CT (120 kV, 80 mA) without contrast was performed for attenuation correction and as an anatomical map. There was no fasting requirement and only a correct hydration previous to both radiotracer administrations was orally promoted.

The acquisition protocol of both PET/CTs included a standard study from skull to proximal legs 5–15 min and 100–120 min after [18F]F-choline and [18F]DCFPyL intravenous administration (activity of 2–4 MBq/Kg and 4–5 MBq/Kg, respectively). We also administrated diuretic medication before any tracer injection. A delayed study of the pelvis, in cases with significant urinary bladder retention or doubtful evaluation, was performed 30–60 min after [18F]F-choline and [18F]DCFPyL standard studies.

### 2.3. Image Analysis and Interpretation

The emission data was corrected for scatter, random coincidence events and system dead time using the provided software. All [18F]F-choline and [18F]DCFPyL scans were evaluated in the Advantage Workstation software version 4.7 (GE Healthcare) allowing review of PET, CT and fused imaging data. Two experienced nuclear medicine physicians evaluated both scans, and a third observer reviewed them in case of discordances. Any focal uptake higher than adjacent background, that did not correspond to physiological uptake, urinary excretion or benign conditions, was considered PET-positive and, thus, probably disease related.

Lesions identified with both tracers were classified in local recurrence (T), lymph nodes (N) and metastases (M), using miTNM stage defined by PROMISE criteria [10].

[18F]F-choline or [18F]DCFPyL avid lesions lacking histopathological verification were rated as malignant if there was a corresponding anatomical finding suspicious for malignancy on MRI or if it was considered clinically malignant in the follow-up by multidisciplinary committee (normalization of the PSA after targeted therapy). Otherwise, these uptakes were considered false positive. In addition, [18F]F-choline avid lesions without any correspondence to [18F]DCFPyL-PET/CT were considered false positive in some cases because of the false positive rate of [18F]F-choline in inflammatory or infectious process demonstrated in the literature [12].

### 2.4. Therapeutic Management and Follow-Up

All diagnostic procedures and treatments undertaken, including biopsies, surgeries, radiotherapy and duration and type of systemic therapy were documented in the follow-up.

Different curative options for BCR are available depending on initial radical treatment. For patients who underwent RP, prostatic fossa radiotherapy is a possibility for either positive or negative prostatic fossa disease detection on PET/CT. Pelvic nodal recurrence can be treated with stereotactic body radiotherapy (SBRT) or surgery. On the other hand, patients who underwent radical prostate radiotherapy have more limitations for a new radiotherapy procedure, in case of prostatic radiotracer uptake on PET/CT, except for brachytherapy, needing a previous histologic confirmation of active disease. Polimetastatic disease (>3) with extension to the retroperitoneal territory and/or bones is treated with systemic therapy (ADT) with/without a combination of androgen receptor-axis-targeted therapies (ARAT) in cases of more extensive disease.

Changes in therapeutic management because of [18F]DCFPyL, compared with [18F]F-choline-PET/CT information, were assessed. We considered that changes in management happened when [18F]DCFPyL-PET/CT modified treatment decision reached after [18F]F-choline findings. Moreover, the added therapeutic impact of [18F]DCFPyL-PET/CT over [18F]F-choline-PET/CT (escalation vs. de-escalation) was assessed. Escalation was defined as locorregional radiotherapy/surgery or ARAT (Abiraterone, Apalutamide or Enzalutamide) in cases of regional or metastatic disease, respectively, only detected by [18F]DCFPyL. De-escalation with only follow-up was decided in cases of a negative [18F]DCFPyL and positive [18F]F-choline, considering that the latter was false positive [12], or when PET/CT results were different and therapeutic decision after [18F]F-choline was

not performed, for example, more disease on [$^{18}$F]DCFPyL-PET/CT that allowed the standard treatment (ADT) instead of local treatment. No therapeutic impact was considered if (i) the results of both scans were concordant or whether different, no differences in treatment were reported compared with the information derived from [$^{18}$F]F-choline, and (ii) patients' clinical conditions did not allow the treatment change planned as a result of [$^{18}$F]DCFPyL-PET/CT.

In follow-up, all diagnostic procedures and treatments undertaken were documented. Serial PSA was obtained every 3 months after planned treatment. Initial treatment response was defined as a drop in PSA levels of greater than 50% from pre-treatment levels at least 6 months after treatment administration. For local curative treatments (surgery or radiotherapy) a minimum time window of 2 months with respect to [$^{18}$F]DCFPyL was considered reliable for assessing efficacy. Men with ADT as part of their treatment were not included in this analysis.

### 2.5. Statistical Analysis

Statistical analysis was performed using SPSS software (v. 28). Quantitative variables were represented by mean and standard deviation and qualitative variables by frequency and percentage. Relation between qualitative variables was studied using Chi-squared Pearson test. Kolmogorov–Smirnov test was used to study normality of the quantitative variables with result of non-normal variables, and the nonparametric tests Kruskal–Wallis and Mann–Whitney were used to compare the means of the quantitative variables. Overall, DR for [$^{18}$F]F-choline and [$^{18}$F]DCFPyL PET/CTs and concordance (Kappa, k) were assessed, classifying the results as poor (<0.20), weak (0.21–0.40), moderate (0.41–0.60), good (0.61–0.80) and very good (0.81–1.00). We also analysed DR of T, N and M recurrence for both tracers, and concordance.

We statistically analysed the correlation between patients' characteristics classified in groups (grade Gleason group, EAU classification, recurrence PSA and kinetics) and PET/CTs results, both DR and miTNM. In all cases, a $p$ value < 0.05 was considered statistically significant. Based on the obtained results, a second assessment was focused on the search of strongest cut-off values of PSA, achieved by receiver operating characteristic (ROC) curve, for the prediction of metastatic disease in comparison with exclusively T and/or N disease.

### 2.6. Search Strategy and Study Selection for the Review

Two authors (LGZ and AMGV) performed a computer literature search on PubMed/MEDLINE databases to find relevant retrospective or prospective published articles on head-to-head comparison between choline-based tracers and PSMA-targeting radiopharmaceuticals in BCR PCa. The exclusion criteria were (i) articles not in English language, (ii) review articles, editorials or letters, comments, conference proceedings, case reports or small case series (<20) and (iii) no head-to-head comparison among these two imaging methods.

The researchers independently reviewed the titles and abstracts of the retrieved articles, applying the inclusion and exclusion criteria. After the selection, the full-text version of the remaining articles, to assess their eligibility for inclusion, were obtained resolving disagreements in a consensus meeting.

## 3. Results

One hundred and thirty-eight patients were enrolled. All the patients' characteristics are presented in Table 1.

**Table 1.** Baseline characteristics of 138 study subjects.

| Characteristic | Value |
|---|---|
| Age (years) | |
| Mean ± SD | 69.77 ± 7.54 |
| Range | 55–87 |
| Grade group | |
| 1 | 46 (33.3%) |
| 2 | 39 (28.3%) |
| 3 | 30 (21.7%) |
| 4 | 12 (8.7%) |
| 5 | 11 (8%) |
| EAU classification (D'Amico risk) | |
| Low | 24 (17.4%) |
| Intermediate | 38 (27.5%) |
| High | 76 (55.1%) |
| Primary treatment | |
| Surgery | 48 (34.8%) |
| Radiotherapy | 60 (43.5%) |
| Both | 30 (21.7%) |
| PSA closest to PET/CTs (ng/mL) | |
| Mean ± SD | 2.80 ± 4.83 |
| PSA ≤ 1 | 46 (33.4%) |
| 1 < PSA ≤ 2 | 17 (12.3%) |
| PSA > 2 | 75 (54.3%) |
| PSAdt (month) | |
| Mean ± SD | 7.34 ± 11.74 |
| ≤6 | 73 (52.9%) |
| >6 | 65 (47.1%) |
| PSAvel (ng/mL/month) | |
| Mean ± SD | 0.26 ± 0.68 |
| ≥0.2 | 45 (32.6%) |
| <0.2 | 93 (67.4%) |
| Biochemical relapse | |
| First | 100 (72.5%) |
| Second or further | 38 (27.5%) |

PSA: prostate-specific antigen; SD: standard deviation; PSAdt: PSA doubling time, PSAvel: PSA velocity; EAU: European Association of Urology.

### 3.1. Detection Rate and TNM Staging by [18F]DCFPyL and [18F]F-choline PET/CT

[18F]DCFPyL showed a higher DR than [18F]F-choline, 64.5% (89/138) and 33.3% (46/138), respectively. Both scans were negative in 44 patients (31.9%) and positive in 41 (29.7%). However, in 20/41 patients, [18F]DCFPyL visualized additional lesions compared with [18F]F-choline, which entailed miTNM stage change in 17 patients (Figure 1).

On the other hand, [18F]DCFPyL was positive alone in 48/89 (53.9%) patients, being oligometastatic in 25 (Figure 2). Five patients were exclusively positive with [18F]F-choline-PET/CT, and thus, [18F]DCFPyL down-staged [18F]F-choline results from positive to negative (3 follow-up, 1 biopsy (negative) and 1 ADT). [18F]DCFPyL up-staged 5/21 patients with oligometastatic disease on [18F]F-choline-PET/CT to polimetastatic disease after [18F]DCFPyL.

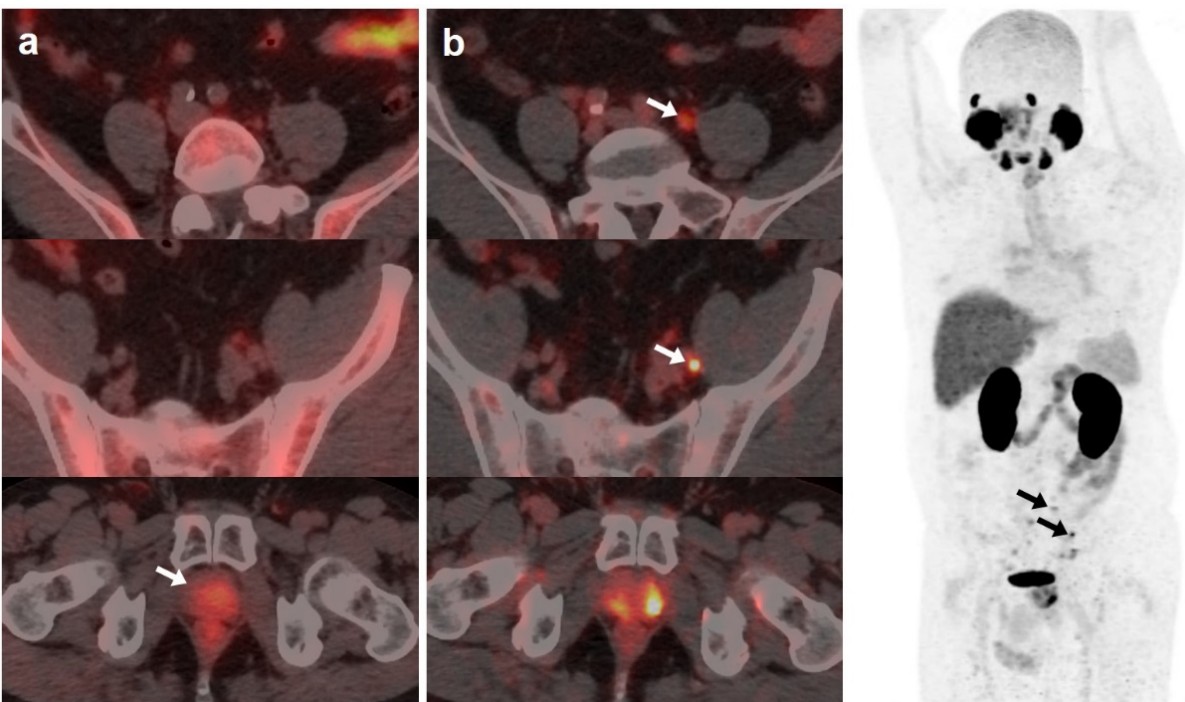

**Figure 1.** 59-year-old patient. Gleason 7 PCa treated with radiotherapy plus ADT. After ADT withdrawal BCR was detected (PSA 2.44 ng/mL, PSAdt 2.6 months, PSAvel 0.15 ng/mL/month). [18F]F-choline (**a**) demonstrated only prostatic uptake (white arrow) and [18F]DCFPyL-PET/CT (**b**) showed prostatic tracer uptake and lymph nodes metastases (white and black arrows). Time window of sixteen days between both scans. [18F]DCFPyL changed therapeutic management allowing escalation (ADT + Apalutamide).

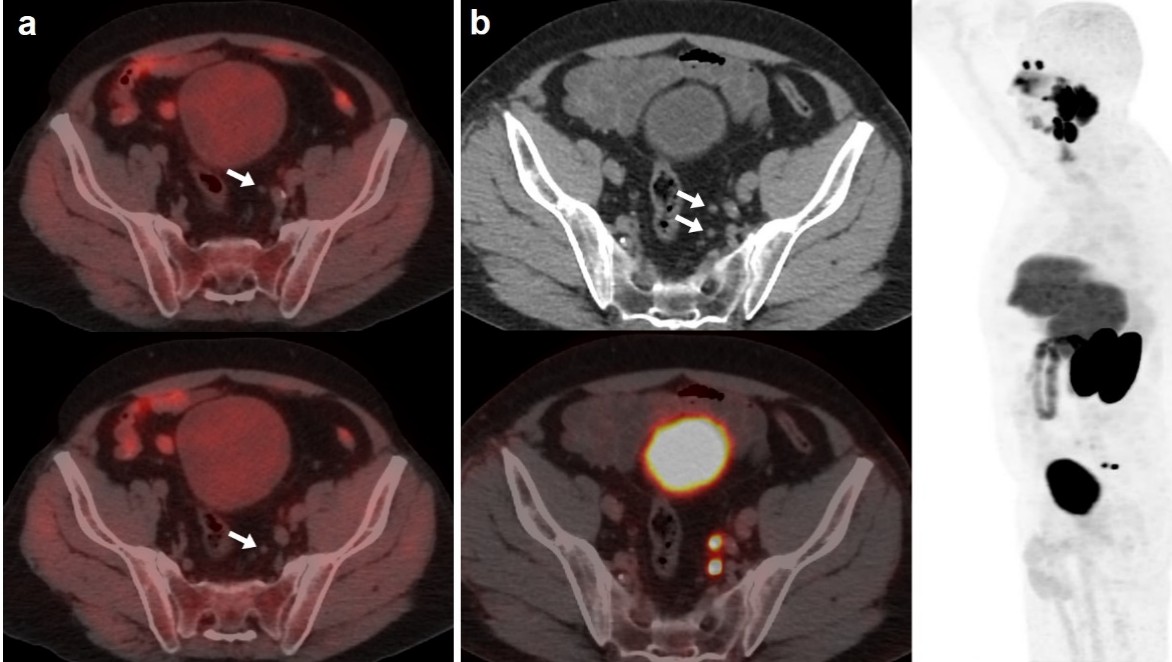

**Figure 2.** 67-year-old patient. Gleason 7 PCa treated with RP. First BCR treated with prostate fossa radiotherapy. Second BCR (PSA 0.63 ng/mL, PSAdt 8.6 months, PSAvel 0.04 ng/mL/month) scanned with [18F]F-choline (**a**) and [18F]DCFPyL PET/CT (**b**), time window of six days. Lymph nodes metastases (white arrows) were demonstrated only on [18F]DCFPyL scan, changing therapeutic management (escalation). Patient underwent lymph nodes SBRT. PSA level decreased.

[$^{18}$F]DCFPyL and [$^{18}$F]F-choline PET/CTs T DR was 33.3% and 19.6%, respectively, with a moderate concordance. N DR was 27.5% and 13.8%, respectively. However, the most significant difference was found for M DR, 30.4% and 8.7%, for [$^{18}$F]DCFPyL and [$^{18}$F]F-choline, respectively (Table 2).

**Table 2.** Per patient miTNM obtained from [$^{18}$F]DCFPyL and [$^{18}$F]F-choline PET/CT, with concordance (kappa).

| | | | **[$^{18}$F]DCFPyL** | | | |
| | | | **(+)** | **(−)** | **Total** | |
|---|---|---|---|---|---|---|
| T | | (+) | 20 | 7 | 27 | |
| | | (−) | 26 | 85 | 111 | |
| | | Total | 46 | 92 | 138 | k = 0.403 (*p* < 0.001) |
| N1 | | (+) | 4 | 8 | 12 | |
| | | (−) | 15 | 111 | 126 | |
| | | Total | 19 | 119 | 138 | k = 0.143 (*p* = 0.086) |
| N2 | [$^{18}$F]F-choline | (+) | 4 | 2 | 6 | |
| | | (−) | 14 | 118 | 132 | |
| | | Total | 18 | 120 | 138 | k = 0.287 (*p* < 0.001) |
| M1a | | (+) | 2 | 1 | 3 | |
| | | (−) | 14 | 121 | 135 | |
| | | Total | 16 | 122 | 138 | k = 0.181 (*p* = 0.003) |
| M1b | | (+) | 5 | 2 | 7 | |
| | | (−) | 16 | 115 | 131 | |
| | | Total | 21 | 117 | 138 | k = 0.304 (*p* < 0.001) |
| M1c | | (+) | 2 | 0 | 2 | |
| | | (−) | 3 | 133 | 136 | |
| | | Total | 5 | 133 | 138 | k = 0.562 (*p* < 0.001) |

T: local recurrence; N1: single lymph node region; N2: multiple lymph node regions (≥2); M1a: extrapelvic lymph nodes; M1b: bone involvement; M1c: other sites; k: kappa.

Regarding first to subsequent BCR, [$^{18}$F]F-choline DR was 35.6% and 27%, respectively, and [$^{18}$F]DCFPyL DR was 61.4% and 72.9%. No statistical differences were found in first to subsequent BCR DR in neither [$^{18}$F]F-choline or [$^{18}$F]DCFPyL (*p* = 0.435 and *p* = 0.164, respectively). We found weak (k = 0.378, *p* < 0.001) and poor (k = 0.079, *p* < 0.467) concordance in first to subsequent BCR DR between [$^{18}$F]F-choline and [$^{18}$F]DCFPyL.

### 3.2. Correlation between PET/CT Results and PSA Kinetics

Both [$^{18}$F]DCFPyL and [$^{18}$F]F-choline PET/CT DR showed significant associations with PSA groups and PSAvel. No significant association was found with PSAdt. [$^{18}$F]DCFPyL DR was 81.3% in patients with PSA > 2 ng/mL, higher than patients with 1 < PSA ≤ 2 (58.8%) and PSA ≤ 1 ng/mL (39.1%). It was also higher in patients with PSAvel > 0.2 ng/mL/month (90.5%) compared with those with PSAvel ≤ 0.2 ng/mL/month (53.7%). [$^{18}$F]F-choline DR was also higher in cases with PSA > 2 ng/mL (46.6%), being 35.3% and 13.04% in patients with 1 < PSA ≤ 2 and PSA ≤ 1 ng/mL, respectively, and 52.4% in cases with PSAvel > 0.2 ng/mL/month versus 26.3% with PSAvel ≤ 0.2 ng/mL/month. In addition, mean PSA was statistically different among patients with T and N recurrence on [$^{18}$F]F-choline-PET/CT (*p* = 0.028). Also, differences in mean PSA and PSAdt were found between N and M (*p* = 0.034) and T and M (*p* = 0.031) metabolic disease, respectively, on [$^{18}$F]DCFPyL-PET/CT (Table 3).

Using predefined cut-off values of PSA and PSA kinetics values, significant association was only found between miTNM and PSA groups on [$^{18}$F]F-choline-PET/CT (*p* = 0.033) and not for PSAdt or PSAvel. No statistical association was found between miTNM, PSA or PSA kinetics on [$^{18}$F]DCFPyL-PET/CT. In ROC analysis, only a PSAdt cut-off of 4.09 months showed significant association for the prediction of M stage with [$^{18}$F]F-choline-PET/CT (66.7% sensitivity, 73.8% specificity, 0.720 AUC, *p* = 0.012). For [$^{18}$F]DCFPyL-PET/CT, the obtained cut-offs in the prediction of M stage: PSA of 2.41 ng/mL (66.7% sensitivity, 64.4%

specificity, 0.675 AUC, *p* = 0.002), PSAdt of 5.59 months (61.1% sensitivity, 60.8% specificity, 0.679 AUC, *p* = 0.001) and PSAvel of 0.13 ng/mL/month (66.7% sensitivity, 61.4% specificity, 0.723 AUC, *p* < 0.001).

**Table 3.** PSA, PSAdt and PSAvel (mean ± SD) in miTNM comparison of [$^{18}$F]DCFPyL and [$^{18}$F]F-choline.

| | | [$^{18}$F]F-choline | [$^{18}$F]DCFPyL |
|---|---|---|---|
| PSA (ng/mL) | T | 3.95 ± 1.92 | 3.17 ± 2.16 |
| | N | 2.68 ± 2.10 | 2.25 ± 2.14 |
| | M | 2.73 ± 1.86 | 4.63 ± 8.67 |
| PSAdt (months) | T | 5.07 ± 12.13 | 7.56 ± 10.83 |
| | N | 6.13 ± 4.23 | 5.87 ± 3.51 |
| | M | 9.32 ± 18.42 | 7.34 ± 11.20 |
| PSAvel (ng/mL/month) | T | 0.45 ± 0.79 | 0.23 ± 0.36 |
| | N | 0.28 ± 0.23 | 0.18 ± 0.15 |
| | M | 0.34 ± 0.44 | 0.56 ± 1.19 |

### 3.3. Therapeutic Impact and Follow-Up

As a result of [$^{18}$F]DCFPyL-PET/CT, therapeutic management was changed in 40/138 (29%) patients compared with [$^{18}$F]F-choline-PET/CT based planning treatment. Escalation was elected in 34 patients: 6 radiotherapy, 5 radiotherapy plus ADT, 6 surgery, 1 prostate cryoablation and 16 ARAT. De-escalation occurred in 6 patients: follow-up in 4 cases (3 [$^{18}$F]DCFPyL negative and 1 with prostatic uptake with both radiotracers and no malignant disease confirmed by biopsy) and ADT instead of local treatment in 2 cases. A potential therapeutic change, derived from [$^{18}$F]DCFPyL-PET/CT information, was not achieved because of patient comorbidities in 11 patients.

Derived from positive [$^{18}$F]DCFPyL-PET/CT, 19 patients underwent additional diagnostic procedures to confirm the results: 8 by imaging (3/8 was confirmed) and 11 by histological analysis (8/11 was confirmed) (Figures 3 and 4).

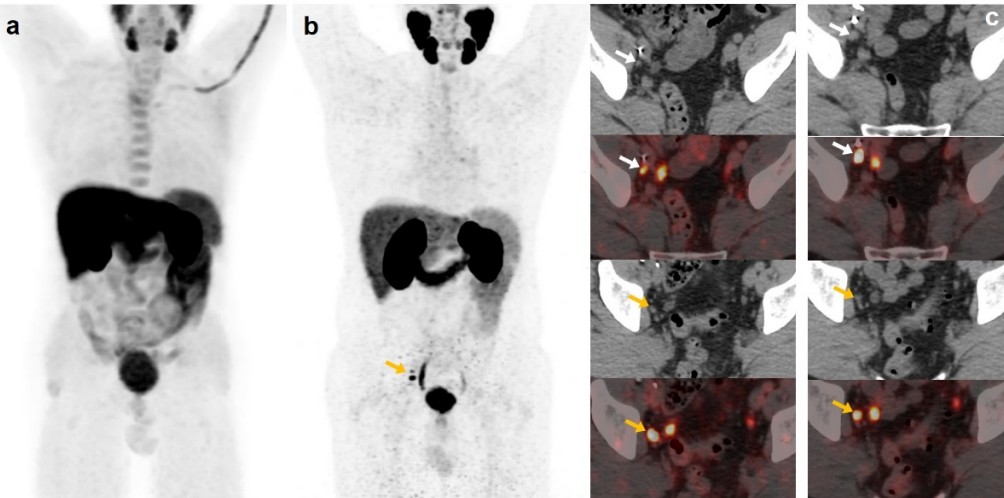

**Figure 3.** 55-year-old patient. Gleason 8 PCa treated with RP. First BCR treated with prostate fossa radiotherapy. Second BCR (PSA: 0.84 ng/mL, PSAdt 5.99 months, PSAvel 0.07 ng/mL/month). [$^{18}$F]F-choline-PET/CT negative (**a**). [$^{18}$F]DCFPyL-PET/CT (**b**), time window of twenty days, revealed two right external iliac lymph nodes metastases (white and yellow arrows). Lymphadenectomy was decided (escalation), without histopathological confirmation of malignancy. In follow-up, PSA progressed (2.07 ng/mL) and an additional [$^{18}$F]DCFPyL-PET/CT (**c**) showed exactly same lymph nodes (white and yellow arrows). SBRT was administered decreasing the PSA level, reclassifying [$^{18}$F]DCFPyL-PET/CT results as true positive.

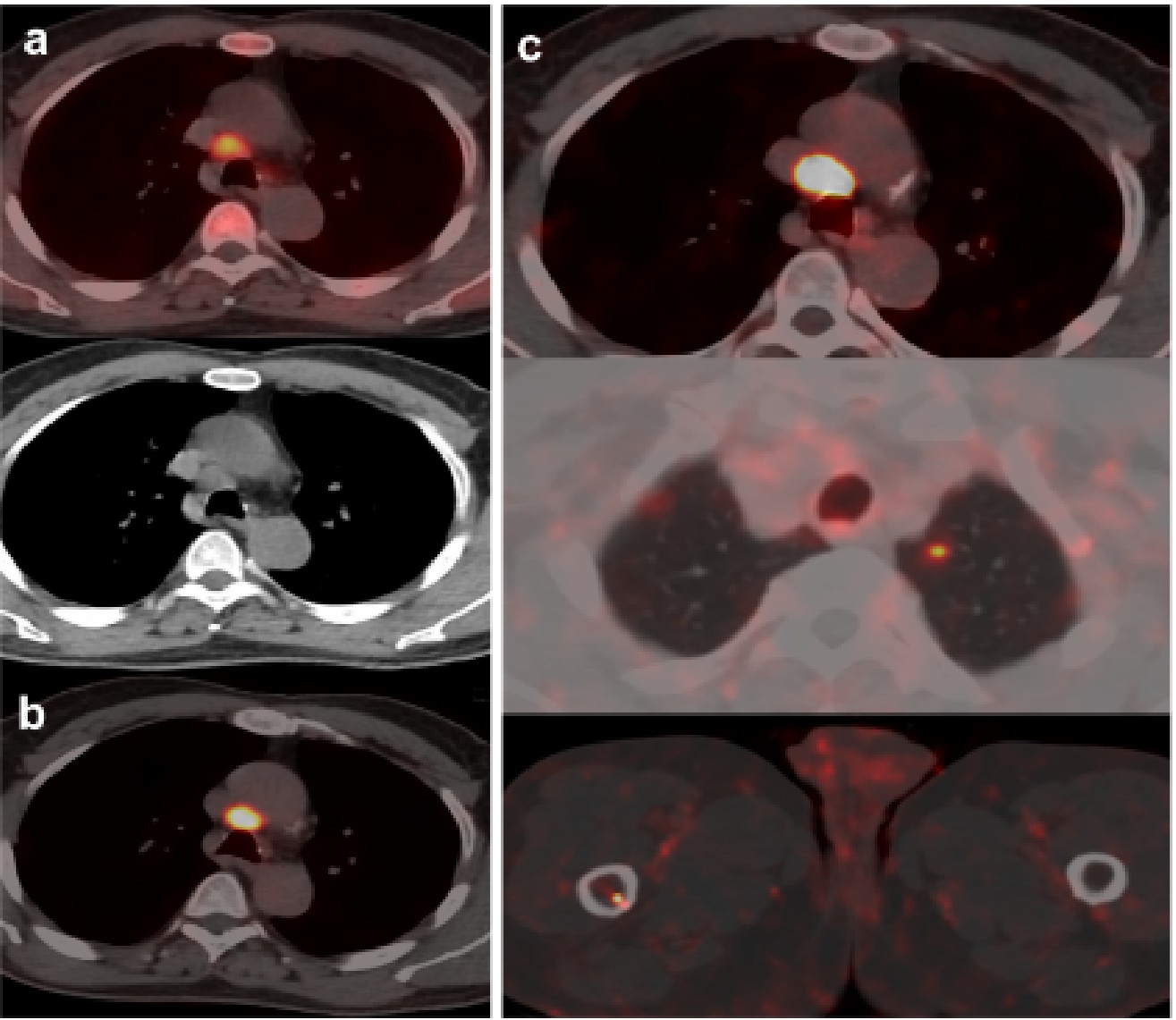

**Figure 4.** 70-year-old patient. Gleason 9 PCa, treated initially with RP and radiotherapy after his first BCR. Second BCR (PSA 0.7 ng/mL, PSAdt 5.6 months, PSAvel 0.05 ng/mL/month) with [18F]F-choline (**a**) and [18F]DCFPyL scans showing mediastinal lymph node tracer uptake (**b**) reported as inflammatory process. Follow-up was decided and PSA level continued increasing. A new [18F]DCFPyL scan (**c**) was performed 3 months later, showing an increase in size and metabolism of mediastinal lymph node with additional microfoci of radiotracer uptake in lung and bone, suspicious of metastases. An endobronchial ultrasound-guided lymph node biopsy confirmed prostatic origin of metastasis. ADT + Apalutamide was initiated (escalation).

[18F]DCFPyL-PET/CT was negative in 49/138 patients (7 low, 14 intermediate and 28 high risk). Follow-up without active treatment was adopted in 29 patients (4 positive [18F]F-choline-PET/CT) and 20 intermediate/high risk patients underwent treatment (12 prostatic fossa radiotherapy, 8 ADT, 1/8 [18F]F-choline-PET/CT positive). Regarding the false positive, six patients with positive [18F]DCFPyL-PET/CT (2 prostate gland, 3 bone and 1 rectum) had a normal MRI (Figures 5 and 6). Ten patients were [18F]F-choline-PET/CT positive and considered false positive (2 prostate gland, 5 lymph nodes, 2 bone, 1 pelvic mass) due to [18F]DCFPyL-PET/CT result, biopsy or clinical follow-up.

For patients who benefited from a treatment change, local treatments were exclusively guided by [18F]DCFPyL in eleven patients. Follow-up showed: PSA decreasing in 6 (4 radiotherapy, 1 cryoablation, 1 surgery), PSA increasing in 2 surgically treated patients (in one patient, an ulterior [18F]DCFPyL-PET/CT revealed an incomplete surgical procedure, and in the other one, surgical procedure was performed almost 7 months after [18F]DCFPyL-PET/CT), in 2 patients biochemical progression occurred before treatment decision and the remaining patient was missed.

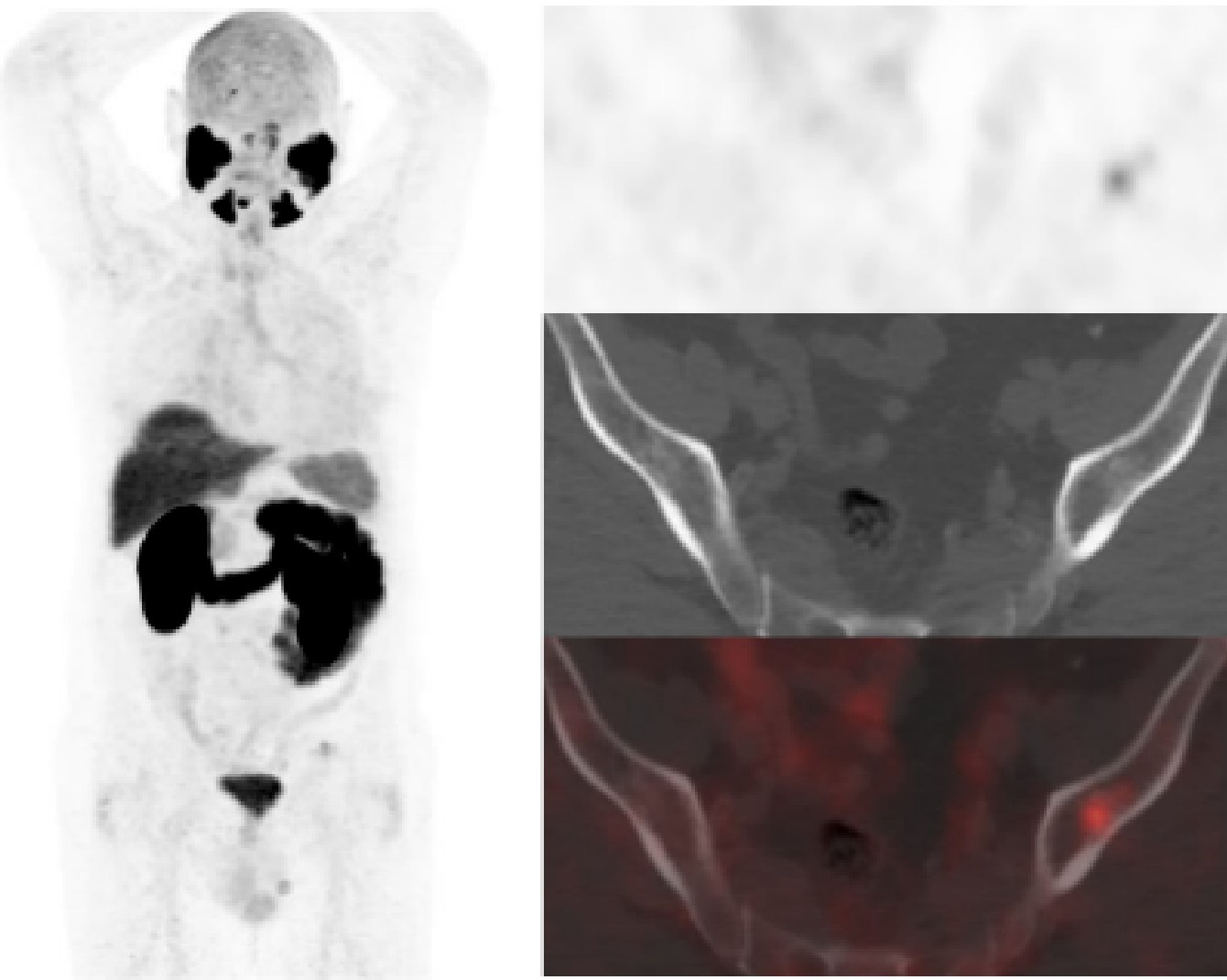

**Figure 5.** BCR in a 71-year-old patient (PSA: 0.26 ng/mL, PSAdt: 1.09 months, PSAvel: 0.2 ng/mL/month) after RP of PCa (Gleason 6, pT2c). [18F]DCFPyL scan showed a slight uptake on left iliac bone with minimal sclerotic changes. Previous negative [18F]F-choline scan (time window of one week). MRI did not confirm malignancy of PSMA uptake (false positive). Prostatic bed radiotherapy was given and PSA level decreased.

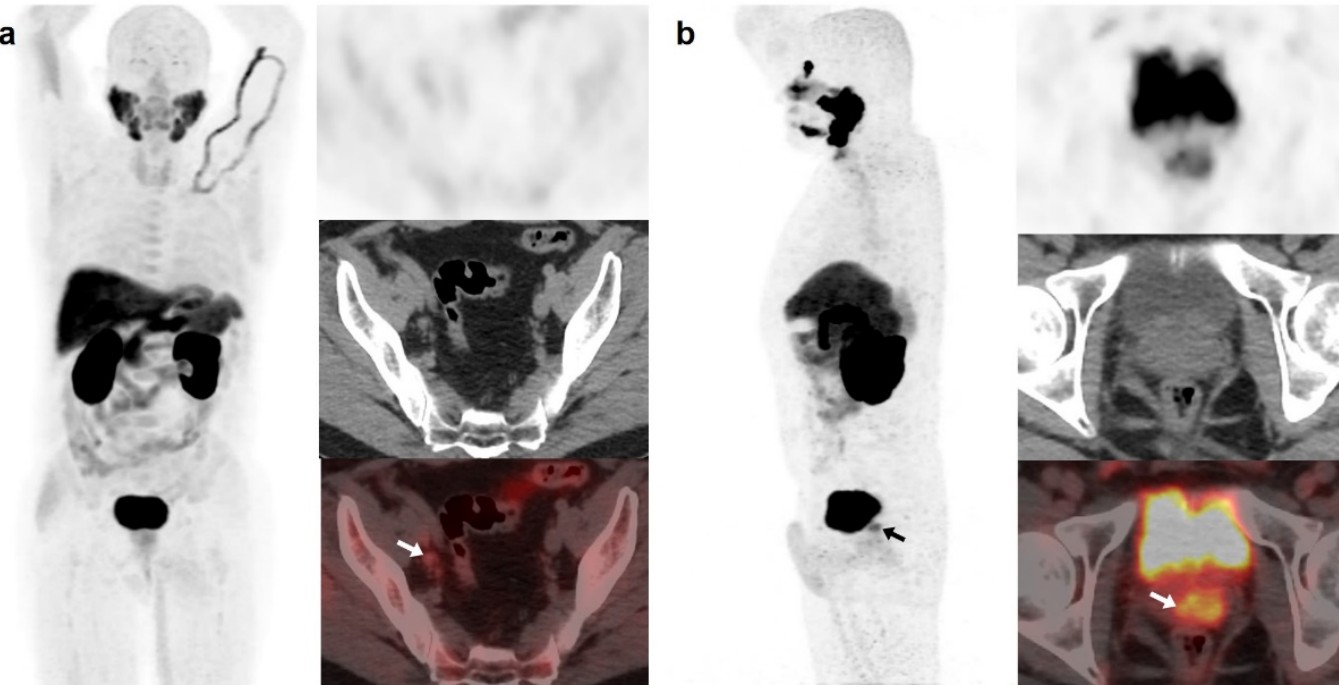

**Figure 6.** 56-year-old patient. PCa (Gleason 6) treated with braquitherapy. BCR (PSA: 5.4 ng/mL, PSAdt 6.17 months, PSAvel 0.55 ng/mL/month). [$^{18}$F]F-choline (**a**) showed prostate gland uptake and right external iliac lymph nodes metastasis (white arrow). One month later, the patient was also scanned with [$^{18}$F]DCFPyL (**b**) revealing only prostate gland pathological tracer uptake (white and black arrow). Prostate biopsy was negative (false positive). Follow-up was decided and PSA level keeps oscillating (4–5 ng/mL) with an additional negative [$^{18}$F]DCFPyL-PET/CT one year later.

### 3.4. Literature Review

Literature search results from PubMed/MEDLINE revealed 17 articles. Reviewing titles and abstracts, seven articles were excluded: four because they were not from the field of interest of this review and three as they were reviews. Ten articles were selected and retrieved in full-text version [7–9,13–19]. Data of 1868 patients with BCR PCa who underwent choline-based tracers and PSMA ligands PET/CT were eligible for the analysis (systematic review). Methodology and results of the selected papers are summarized on Table 4.

Table 4. Summary of studies comparing choline-labelled tracers and PSMA-targeting radiopharmaceuticals PET/CT in BR of PCa.

| Author, Year, Study Type | n | Requirements for Requesting PET/CT with PSMA-Targeting Tracers. PSA Values (Median/Mean ± SD) | PET Radiotracers/Time Interval between Studies (Median/Mean ± SD/Range) | Diagnostic and Therapeutic Impact |
|---|---|---|---|---|
| Chevalme, 2021, R [7] | 1084 | Previous [18F]F-choline-PET/CT negative (924) or equivocal (160). Median PSA 1.7 ng/mL | [68Ga]Ga-PSMA-11 vs. [18F]F-choline/Median 42 days (4–100) | [68Ga]Ga-PSMA-11 was positive in 62% and in 82% of previous [18F]F-choline negative or equivocal, respectively. Overall DR of 68%. No therapeutic impact assessment. |
| Barbaud, 2019, R [8] | 42 | Previous [18F]F-choline-PET/CT negative (26) or doubtful. Mean PSA, PSAdt and PSAvel of 4.1 ± 5.1 ng/mL, 8.5 ± 7.4 months and 4 ± 4.8 ng/mL/y, respectively | [68Ga]Ga-PSMA-11 vs. [18F]F-choline/Median 41 days (14–243) | DR for [68Ga]Ga-PSMA-1: 80.9% (PB: 19 p, LN: 18p, M1b: 8p, M1c: 3p). Change in therapeutic management in 73.8%. |
| Witkowska-Patena, 2019, P [9] | 40 | None (90% negative or equivocal [18F]F-choline)/Mean PSA 0.77 ± 0.61 ng/mL | [18F]PSMA-1007 vs. [18F]F-choline/Mean 54 ± 21 days. Median 58 (12–105) d | DR for [18F]PSMA-1007 and [18F]F-choline of 60% and 5%, respectively. [18F]PSMA-1007 detected more lesions. In 70% of scans, [18F]PSMA-1007 upgrades [18F]F-choline result, from negative to positive. No therapeutic impact assessment. |
| Alonso, 2018, P [13] | 36 (a) | None/Median PSA 3.3 ng/mL | [68Ga]Ga-PSMA-11 vs. [11C]-choline/1–2 weeks | DR for [68Ga]Ga-PSMA-11 and [11C]-choline of 75% and 53%, respectively. [68Ga]Ga-PSMA-11 detected more lesions. |
| Cantiello, 2018, R [14] | 43 | None/Median PSA, PSAdt and PSAvel of 0.8 ng/mL, 4 months and 2.6 ng/mL/y, respectively | [64Cu]PSMA-617 vs. [18F]F-choline/Median 2.2 weeks (1–3) | DR for [64Cu]PSMA-617 and [18F]F-choline of 74.4% and 44.2%, respectively. PB 30.2%, N1a ± PB: 9/43, M1b ± PB: 7/43. No therapeutic impact assessment. |
| Caroli, 2018, P [15] | 314 | Previous [18F]F-choline-PET/CT negative or dubious. Median PSA 0.83 ng/mL | [68Ga]Ga-PSMA-11 vs. [18F]F-choline/< 30 days | DR of 62.7% for [68Ga]Ga-PSMA-11 (67% in 88 patients with negative [18F]F-choline PET/CT). No therapeutic impact assessment. |
| Schwenck, 2017, R [16] | 123 (b) | None. Median PSA and PSAdt of 2.7 ng/mL and 4 months, respectively | [68Ga]Ga-PSMA-11 vs. [11C]-choline/< 24 h | DR for [68Ga]Ga-PSMA-11 and [11C]-choline of 83% and 79%, respectively, in biochemical relapse (103 p). No therapeutic impact assessment. |
| Bluemel, 2016, R [17] | 125 (c) | A previous [18F]F-choline-PET/CT negative in 41 patients. Mean PSA, PSAdt and PSAvel of 5.4 ± 12.73 ng/mL, 9.9 ± 10.6 months and 7 ± 25 ng/mL/y, respectively | [68Ga]Ga-PSMA-I&T vs. [18F]F-choline/Mean 19 ± 16 days | [68Ga]Ga-PSMA-I&T detected disease in 43.8% of patients with a previous negative [18F]F-choline. No therapeutic impact assessment. |

**Table 4.** *Cont.*

| Author, Year, Study Type | n | Requirements for Requesting PET/CT with PSMA-Targeting Tracers. PSA Values (Median/Mean ± SD) | PET Radiotracers/Time Interval between Studies (Median/Mean ± SD/Range) | Diagnostic and Therapeutic Impact |
|---|---|---|---|---|
| Morigi, 2015, P [18] | 38 | None/Mean PSA and PSAdt of 1.72 ± 2.54 ng/mL and 15.6 ± 22.1 months, respectively | [68Ga]Ga-PSMA-11 vs. [18F]F-choline/<30 days | DR for [68Ga]Ga-PSMA-11 and [18F]F-choline of 66% and 32%, respectively. [68Ga]Ga-PSMA-11 detected more lesions. Change in therapeutic management in 63%. |
| Afshar-Oromieh, 2014, R [19] | 37 | None. Mean PSA 11.1 ± 24.1 ng/mL | [68Ga]Ga-PSMA-11 vs. [18F]F-choline/Mean 12.1 ± 8.4 days | DR for [68Ga]Ga-PSMA-11 and [18F]F-choline of 86.5% and 70.3%, respectively. PSMA detected more lesions. No therapeutic impact assessment. |

n: number of patients, P: prospective, R: retrospective, PB: prostate bed, LN: lymph nodes, M1b: bone metastases, M1c: visceral metastases, p: patients, DR: detection rate, (a) some not specified patients with androgen deprivation therapy, (b) primary and recurrent prostate cancer, (c) only use PET/CT with PSMA-targeting tracers in patients with a previous negative [18F]F-choline-PET/CT, PSAdt: PSA doubling time, PSAvel: PSA velocity, SD: standard deviation.

## 4. Discussion

In BCR, diagnostic impact of PSMA-targeting radiopharmaceuticals against choline-based tracers is significantly higher in patients with low PSA levels and previous negative/doubtful PET/CT with choline-based tracers in whom the detection of oligometastatic disease might enable metastasis-directed treatments [20,21]. In our study, patients with previous negative/oligometastatic [18F]F-choline-PET/CT were referred to [18F]DCFPyL expecting a benefit by the detection of more metastases. In fact, we found that 5/21 patients with oligometastatic disease in [18F]F-choline-PET/CT were up-staged to polimetastatic after [18F]DCFPyL, similar to previous reported results [16]. The absence of consensus about oligometastatic definition can limit diagnostic impact comparison of different radiotracers. In previous studies, oligometastatic disease was defined as M stage with $\leq 5$ lesions [16,17], whereas other authors considered $\leq 3$ as we did in this study [7]. Chevalme et al. found, using [68Ga]Ga-PSMA-11, oligometastatic disease (1–3 foci) in 31% of the cases with previous negative/doubtful [18F]F-choline [7]. In our sample, [18F]DCFPyL-PET/CT detected oligometastatic disease in the 52.1% (25/48) of choline-negative cases. In a relevant number of cases, we detected positive lymph nodes and bone lesions that showed divergent findings with both tracers. The majority were only [18F]DCFPyL-positive, probably due to higher lesion/background ratio and sensitivity compared with [18F]F-choline that enables detection of smaller lesions.

Regarding the DR, our results are in accordance with previous works, with a [18F]DCFPyL DR of 53.9% in patients with negative/equivocal [18F]F-choline [7,8,15,17]. In addition, we did not observe differences in DR of first BCR with respect to the following, which is contrary to the study of Chevalme et al. [7] that reported a DR lower in first BCR versus previous (63 vs. 72%).

With respect to the influence of PSA kinetics, despite a PSAdt $\leq 6$ months has been reported as a strong predictor of positivity of PET/CT with choline-based tracers [21], we did not find a significant association with DR or miTNM for any of the studied radiotracers with their counterparts. This absence of significant association with pre-defined unfavourable PSA kinetics promotes the interest in exploring other clinical, metabolic and laboratory parameters. In fact, we found that different cut-off values of PSA kinetics were able to predict M stage, especially for [18F]DCFPyL-PET/CT, although with a moderate accuracy.

Regarding disease location on BCR PCa, [18F]DCFPyL-PET/CT detected T in 33.3% of cases, similar to 26% reported by Chevalme et al. [7], but higher than 11% of Barbaud et al. [8], probably explained by higher rate of patients included with RP (76%). We observed higher T detection than [18F]F-choline, although with a moderate concordance (k = 0.403, *p* < 0.001). Discrimination between benign and malignant intraprostatic tissue is hampered by low specificity of choline-based tracers based on high affinity of this radiotracer by benign hyperplasia [22,23]. Lymph node is the most prevalent disease location in BCR PCa, showing PSMA-targeting radiopharmaceuticals with a DR from 34% to 39% in patients with a previous choline-negative [7,8,16]. Our N disease detection using [18F]DCFPyL was lower (27.5%), with a weak concordance with [18F]F-choline. Previous authors found that 55% of the detected lymph nodes were identified with both tracers. Thus, using PSMA-ligands, increase in DR affecting both the number and locations of lymph nodes is a fact [16]. However, the most significant difference in our sample was M detection, 30.4% and 8.7% for [18F]DCFPyL and [18F]F-choline, respectively, in accordance with previous studies [7,16].

With respect to the discordances between both radiotracers, usually PSMA-targeting radiopharmaceuticals spot all choline-positive lesions, and discordances are mainly related to choline-negative/PSMA-positive findings [14,16]. The explanations of these discordances are contradictory and based on: (i) different metastasis environment with a loss of expression of PSMA that can occur in less than 10% of primary or metastatic prostate tumours [24]; (ii) tumour progression between scans in cases of a wide time interval [9]; and (iii) unspecific inflammation that promotes choline uptake in lymph nodes and can explain

additionally choline-positive lymph nodes [25]. The higher PSMA-targeting radiopharmaceutical specificity with respect to choline-based tracers, explained by the overexpression of PSMA glycoprotein, seems more characteristic for PCa than up-regulation of choline kinase [26,27] and could support our consideration of PSMA-targeting tracer findings as standard, as previous authors did [16,18].

Prostatic fossa salvage radiation treatment (SRT) is the current standard of care in men with their first BCR after RP [28]. However, in patients with a previous radical radiotherapy, only brachytherapy is indicated if malignancy is histopathologically confirmed. On the other hand, a second BCR in patients that have undergone previous radiotherapy of prostatic fossa, with or without pelvic lymph nodes involved, reduces the potential local treatment options. Therefore, ADT becomes a real therapeutic option both in patients without located disease or with local disease but with no indication of an additional local treatment (SRT or surgery).

Therapeutic impact of PSMA-targeting radiopharmaceuticals compared with PET/CT with choline-based tracers has been scarcely analysed, ranging from 54% to 74% [8,18]. We observed an impact management in 29% of cases although it could be raised to 37% if patients with comorbidities limited the previously indicated treatment and could have been included. Escalation was considered when the treatment modification involved changing/adding radiotherapy fields or adding ARAT to the systemic ADT. Thus, men with a previous RP without disease or with disease confined to the prostatic fossa on PET/CT imaging were expected to proceed to SRT or a combination of radiotherapy and ADT if few lesions were defined on [18F]DCFPyL-PET/CT or ADT plus ARAT in case of multiple locations (M stage) in patients with no chemotherapy indication. The therapeutic impact derived from [18F]DCFPyL over [18F]F-choline findings allowed treatment escalation in most of our patients (34/40). However, the assessment of therapeutic impact is challenging, being not only dependent of the accuracy of diagnostic techniques but on other factors as previously received treatments and the comorbidities/physical status of the patients, and although [18F]DCFPyL-PET/CT result could have changed the therapeutic management of some patients, this decision was not carried out because of their clinical situation. Diagnostic escalation (additional diagnostic imaging vs. biopsy) to confirm [18F]DCFPyL results is another relevant aspect to be assessed. In our sample, 19 patients underwent additional diagnostic procedures, only 11 by histological analysis, lower than the 24% reported by Morigi et al. [18].

Elective prostatic bed radiotherapy is a controversial issue. About half of men who experience BCR after RP and undergo SRT with the prostatic bed, even when there are no significant imaging findings, are currently cured [29], suggesting that SRT should still be considered despite a negative imaging result [30]. On the other hand, focused radiotherapy based on PET/CT with PSMA-targeting tracers exhibits higher response rates compared with the conventional procedure without metabolic guide, although cannot guarantee undetectable PSA in all the cases and that means PET/CT with PSMA-targeting tracers still underestimate the extent of the recurrent disease [8,30]. In the present analysis, 12 out of 49 patients with a negative [18F]DCFPyL-PET/CT underwent prostatic fossa radiotherapy.

Thus, therapeutic implications derived from the use of PSMA-targeting radiopharmaceuticals can be significant. Target missed in BCR due to insufficient diagnostic work-up may lead to inadequate definition of local disease and to untreated microscopic or macroscopic disease distant from prostatic fossa (N1/M1). The expected result, derived from an earlier and more accurate diagnosis of PET/CT with PSMA-targeting radiopharmaceuticals, is the opportunity for focused therapies, with a reduction in the introduction of ADT and thus the time to ADT-resistance [31,32].

Regarding limitations, histopathological confirmation of our PET/CT results was not always feasible, although it is a controversial issue and probably neither indicated nor ethical. In addition, 2-month period used between both PET/CTs could limit a reliable comparison between both radiotracers in cases of a highly proliferative disease.

However, PCa usually presents with slow growth, and noticeable changes within this period are very unlikely.

With respect to the strengths, this is the first reported experience of the therapeutic impact of [18F]DCFPyL in connection with [18F]F-choline, in parallel comparison, in a significant sample of patients with BCR PCa.

## 5. Conclusions

[18F]DCFPyL provided a higher DR than [18F]F-choline in restaging of BCR, especially in patients with high PSA and unfavourable PSA kinetics, being superior in miTNM staging and showing a fair agreement to [18F]F-choline-PET/CT. Information derived from [18F]DCFPyL changed therapeutic management in a significant number of patients (29%) compared with [18F]F-choline-PET/CT.

**Author Contributions:** Conceptualization, L.G.-Z. and A.M.G.-V.; methodology, L.G.-Z. and A.M.G.-V.; formal analysis, M.A.-S.; investigation, L.G.-Z. and A.M.G.-V.; writing—original draft preparation, L.G.-Z. and A.M.G.-V.; writing—review and editing, L.G.-Z. and A.M.G.-V.; visualization, L.G.-Z. and C.L.-L.; supervision, Á.M.S.-C. All authors have read and agreed to the published version of the manuscript.

**Funding:** This research received no external funding.

**Institutional Review Board Statement:** The study was conducted in accordance with the Declaration of Helsinki and approved by the reference Ethical Committee (internal code of 2022-53; 24 May 2022) of GAI Albacete.

**Informed Consent Statement:** Informed consent was obtained from all subjects involved in the study.

**Data Availability Statement:** Data supporting reported results are only available for local investigators.

**Conflicts of Interest:** The authors have no relevant financial or non-financial interests to disclose.

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
