# Peer review of "Head-to-Head Comparison of [18F]F-choline and Imaging of Prostate-Specific Membrane Antigen, Using [18F]DCFPyL PET/CT, in Patients with Biochemical Recurrence of Prostate Cancer"

_curroncol, doi:10.3390/curroncol30070464_

Round 1

Reviewer 1 Report

Very nice paper, basically confirming already published results saying that studied second generation PSMA PET provides higher DR a superior miTNM staging in recurrent prostate cancer compared to cholin 18F PET.

The conclusion is clear. However, as for a clinical oncologist, being practically a laic in advanced radiology, reading through the results seems to be quite difficult and sometimes confusing with too many outcomes and numbers. Please, try to make the text more smooth and getting a clear point. Also improve tables 2+4 - better description, p values?- not only comments in the  too much condensed text.

I have also some concerns regarding image analysis and interpretation. It may look like evaluating the results of PSMA PET are somewhat favoured over cholin PET. Positive finding on cholin PET is regarded as false positive in case there is no histopathological or imaging confirmation. Contrary to that, PSMA PET is regarded as true positive based on follow-up judged by a multidisciplinary team. Moreover, it can happen that if both are positive and there is no confirmation of the result, cholin PET may be regarded as false positive and PSMA PET as true positive. This may look as a methodological bias to someone not trained in nuclear medicine. Keep in mind that you are going to publish in "Current oncology" read mainly by oncologists. Thank you

a language correction and editing is advisable 

Author Response

Very nice paper, basically confirming already published results saying that studied second generation PSMA PET provides higher DR a superior miTNM staging in recurrent prostate cancer compared to cholin 18F PET. 
The conclusion is clear. However, as for a clinical oncologist, being practically a laic in advanced radiology, reading through the results seems to be quite difficult and sometimes confusing with too many outcomes and numbers. Please, try to make the text more smooth and getting a clear point. Also improve tables 2+4 - better description, p values?- not only comments in the  too much condensed text.

-    We have done some changes in the section “Image analysis and interpretation” as it was noted (underlined in the document). 
-    The tables have also been modified for a better understanding. Tables 2 and 3 were combined (underlined in the document). 

I have also some concerns regarding image analysis and interpretation. It may look like evaluating the results of PSMA PET are somewhat favoured over cholin PET. Positive finding on cholin PET is regarded as false positive in case there is no histopathological or imaging confirmation. Contrary to that, PSMA PET is regarded as true positive based on follow-up judged by a multidisciplinary team. Moreover, it can happen that if both are positive and there is no confirmation of the result, cholin PET may be regarded as false positive and PSMA PET as true positive. This may look as a methodological bias to someone not trained in nuclear medicine. Keep in mind that you are going to publish in "Current oncology" read mainly by oncologists. Thank you
- We understand your concerns regarding the false positive choline results in cases of negative PSMA PET. Previous published literature defines the higher accuracy of PSMA PET with respect to choline PET. So, we based on our FP choline consideration on that. In cases of concordant positive results for both radiotracers, choline was considered as true positive.

Reviewer 2 Report

The authors describe an interesting head to head comparison between F-choline and DCFPyL, although I think that the time of F-Choline in prostate cancer is already behind us. This had been obvious from many studies with Ga-68 and F-18 based PSMA-receptor ligands. Although it might still be a reality in some countries to perform F-choline PET due to reimbursement issues, the evidence already evolved.

The authors are forced to make a complicated construct of the ground truth concerning metastatic lesions identified on the scan were they ended up in party circular reasoning were a lesion can be a true positive if the investigated test is positive and was deemed malignant in a multi-disciplinary tumour board. But how did the tumour board decide? If the lesion is gone after treatment then it is tumour. What is it persists, does that mean it is something else? How is response in other lesions and heterogeneity in response evaluated? When I want to go this way it get complicated. The detection rate is much less prone to al kinds of bias caused by interpretation of response, lack of response and treatment decisions that influence how you determine the ground truth.

In the image  analysis and interpretation the PET reconstruction is discussed, I suppose you also did attenuation correction, but it was not mentioned.

The timings of the scans after injection is not clear. It would be better to describe these separately

In the results section the …( detection rate?) stratified for T N M is describe with reference to table 3. But the data in table 3 contains concordance between the two scan in kappa values??? I think table 2 was meant? But the data in the text contain percentages, the table is in absolute values???

The caption of Table 4 contains sensitivity and specificity data. I have a real issue with the sensitivity calculations here. How are false negative defined in the study?

Concerning the moderate accuracy of prediction of scan positivity with serum parameters, the author may also discuss the expected percentage tumours not expression the target at all

Concerning language:

“Patients with BCR of PCa after radical treatment (RP, radiotherapy, or both) were derived from different hospitals of our region…”. Derived replace with recruited.

“…Spanish Agency of Medication and Health Care Products and after being approved by a multidisciplinary committee and previous patient informed and signed consent.” Discard the word previous.

“The inclusion criteria of selected patients for the present analysis were…” discard of selected patients?.

“Cases when no therapeutic impact happened due to patients’ clinical conditions were recorded” if it is not already clear from the text above that the three categories in which changes in therapeutic  management was scored were “escalation”, “de-escalation” or “unchanged“, then rephrase: if neither escalation or de-escalation was decided it was recorded as unchanged.

Be careful with abbreviations, OD in US-English commonly means overdose.

I would not describe a scan a doubtful, I would rather say ambiguous.

Author Response

Reviewer 2
The authors describe an interesting head to head comparison between F-choline and DCFPyL, although I think that the time of F-Choline in prostate cancer is already behind us. This had been obvious from many studies with Ga-68 and F-18 based PSMA-receptor ligands. Although it might still be a reality in some countries to perform F-choline PET due to reimbursement issues, the evidence already evolved.
The authors are forced to make a complicated construct of the ground truth concerning metastatic lesions identified on the scan were they ended up in party circular reasoning were a lesion can be a true positive if the investigated test is positive and was deemed malignant in a multi-disciplinary tumour board. But how did the tumour board decide? If the lesion is gone after treatment then it is tumour. What is it persists, does that mean it is something else? How is response in other lesions and heterogeneity in response evaluated? When I want to go this way it get complicated. The detection rate is much less prone to al kinds of bias caused by interpretation of response, lack of response and treatment decisions that influence how you determine the ground truth.

In the image analysis and interpretation the PET reconstruction is discussed, I suppose you also did attenuation correction, but it was not mentioned.
-    Yes, attenuation correction was mentioned in “acquisition protocol”: “Low dose CT (120 kV, 80 mA) without contrast was performed for attenuation correction and as an anatomical map”. 

The timings of the scans after injection is not clear. It would be better to describe these separately. 
-    We have made changes in “acquisition protocol” and “image analysis and interpretation” as it was noted (underlined in the document). 

In the results section the …( detection rate?) stratified for T N M is describe with reference to table 3. But the data in table 3 contains concordance between the two scan in kappa values??? I think table 2 was meant? But the data in the text contain percentages, the table is in absolute values???
-    The tables have also been modified for a better understanding (Tables 2 and 3 were combined).  

The caption of Table 4 contains sensitivity and specificity data. I have a real issue with the sensitivity calculations here. How are false negative defined in the study?
Concerning the moderate accuracy of prediction of scan positivity with serum parameters, the author may also discuss the expected percentage tumours not expression the target at all
-    A more detailed explanation has been added in methods and discussion, correcting the previously defined descriptions in order to a better understanding.

Comments on the Quality of English Language
Concerning language:
“Patients with BCR of PCa after radical treatment (RP, radiotherapy, or both) were derived from different hospitals of our region…”. Derived replace with recruited. 
“…Spanish Agency of Medication and Health Care Products and after being approved by a multidisciplinary committee and previous patient informed and signed consent.” Discard the word previous. 
“The inclusion criteria of selected patients for the present analysis were…” discard of selected patients?.
“Cases when no therapeutic impact happened due to patients’ clinical conditions were recorded” if it is not already clear from the text above that the three categories in which changes in therapeutic management was scored were “escalation”, “de-escalation” or “unchanged“, then rephrase: if neither escalation or de-escalation was decided it was recorded as unchanged.
Be careful with abbreviations, OD in US-English commonly means overdose. 
I would not describe a scan a doubtful, I would rather say ambiguous. 

Response: 
-    We have made the changes purposed for the language. 
-    In addition, we have rephrased the “therapeutic impact” paragraph.  
-    A deep review with many changes has been performed in order to do a more direct description and redaction.
-    Some corrections have been introduced on tables. A revised version and clean version (final) are attached in order to locate all the changed done. 

Reviewer 3 Report

The paper is a head-to-head comparison of the performances of 18F-FCH and 18F-DCFPyL in patients with prostate cancer who experience biochemical recurrence .

The manuscript may be interesting but needs some improvements before being published. First of all, the layout of paragraphs and sub-paragraphs should be edited; the same is true for the images and the parts of the text near them.

Secondly, I think it is not necessary to explicit the research strategy, as it is not a review; Discussion Section should be modified, as now it looks more like a review than a discussion on your study itself.

 Moderate editing of English language required

Author Response

Reviewer 3
The paper is a head-to-head comparison of the performances of 18F-FCH and 18F-DCFPyL in patients with prostate cancer who experience biochemical recurrence.
The manuscript may be interesting but needs some improvements before being published. First of all, the layout of paragraphs and sub-paragraphs should be edited; the same is true for the images and the parts of the text near them.
Secondly, I think it is not necessary to explicit the research strategy, as it is not a review; Discussion Section should be modified, as now it looks more like a review than a discussion on your study itself.

Response:
-    We have modified the layout of paragraphs and sub-paragraphs for a better understanding.  
-    The discussion has been extensively reviewed checking that all the issues are present- work related.

Round 2

Reviewer 2 Report

I would like to thank the authors for the provided answers. 

Author Response

Thanks a lot for your comments and review

Reviewer 3 Report

The manuscript has been improved now, but still requires some modifications. First of all, please re-write all the radiopharmaceuticals according to the latest EANM Guidelines (e.s. [18F]...). Secondly, figure in page 6 maybe should be deleted; figures 3, 4 and 6 should be improved (e.g. the arrows are too small); figure in page 11 does not have a caption, please modify it.

Minor editing of English language required

Author Response

The manuscript has been improved now, but still requires some modifications. First of all, please re-write all the radiopharmaceuticals according to the latest EANM Guidelines (e.s. [18F]...). Secondly, figure in page 6 maybe should be deleted; figures 3, 4 and 6 should be improved (e.g. the arrows are too small); figure in page 11 does not have a caption, please modify it.

Response: All the radiopharmaceuticals have been re-written according to the latest EANM Guidelines (underlined in the document). In addition, figures have been improved as it was noted.